# Dihydroquinazolinones as adaptative C($sp^3$) handles in arylations and alkylations via dual catalytic C–C bond-functionalization

Xin-Yang Lv[1,2], Roman Abrams[1] & Ruben Martin [1,2,3✉]

C–C bond forming cross-couplings are convenient technologies for the construction of functional molecules. Consequently, there is continual interest in approaches that can render traditionally inert functionality as cross-coupling partners, included in this are ketones which are widely-available commodity chemicals and easy to install synthetic handles. Herein, we describe a dual catalytic strategy that utilizes dihydroquinazolinones derived from ketone congeners as adaptative one-electron handles for forging C($sp^3$) architectures via α C–C cleavage with aryl and alkyl bromides. Our approach is achieved by combining the flexibility and modularity of nickel catalysis with the propensity of photoredox events for generating open-shell reaction intermediates. This method is distinguished by its wide scope and broad application profile––including chemical diversification of advanced intermediates––, providing a catalytic technique complementary to existing C($sp^3$) cross-coupling reactions that operates within the C–C bond-functionalization arena.

[1] Institute of Chemical Research of Catalonia (ICIQ), The Barcelona Institute of Science and Technology, Av. Països Catalans 16, 43007 Tarragona, Spain. [2] Departament de Química Analítica i Química Orgànica, Universitat Rovira i Virgili, c/Marcel·lí Domingo, 1, 43007 Tarragona, Spain. [3] ICREA, Passeig Lluís Companys, 23, 08010 Barcelona, Spain. ✉email: rmartinromo@iciq.es

Transition-metal-catalyzed cross-coupling reactions of nucleophilic and electrophilic components are powerful methods for rapidly forming carbon-carbon bonds[1–3]. These approaches have been widely applied to the preparation of biologically-relevant molecules and functional materials, by academic and industrial institutions alike[4–7]. Consequently, great interest exists for the development of new cross-coupling synthons that operate under ambient conditions, as this increases the structural diversity of accessible molecules within drug discovery programs[8–12]. The broad utility of ketones as chemical precursors[13,14], the plethora of methods for their preparation[15–17], and their prevalence as medicinal and commodity chemicals make them ideal targets for chemical innovation[18–21]. Synthetic manipulations of ketones generally rely on their latent polarity, specifically the electrophilicity of C=O bonds and nucleophilicity of enolate related structures (Fig. 1a, path a)[22,23]. In sharp contrast, the selective and catalytic cleavage of ketone α C−C bonds as a platform for installing chemical functionality still remains challenging (Fig. 1a, path b). However, such techniques hold promise for creating conceptually new disconnections during retrosynthetic analysis and methods towards otherwise inaccessible compounds[24]. Traditional methods for inserting a single atom into the α C–C bond of ketones include the venerable Büchner−Curtius−Schlotterbeck and Baeyer−Villiger reactions (Fig. 1b)[25,26]. More recently, significant interest has been directed at using transition metal catalysis to achieve α C−C cleavage of ketones followed by C−C bond-forming reactions[27–37]. These approaches are generally specific to either strained motifs[27–31] or 1,3-dicarbonyl substrates[32,33], require directing group activation[34,35], use high-temperatures[36,37], or a combination of the preceding (Fig. 1c). This is presumably due to the directionality and high C−C bond-strength of ketones, thus making activation challenging. Consequently, new strategies are still required to fully realize ketones or derivatives thereof as traceless handles in cross-coupling reactions.

Metallaphotoredox catalysis has gained momentum as a powerful synthetic tool[38–40], in particular by allowing alcohols[41–43], primary amines[44,45] and aldehydes[46–48] to be used as adaptative $C(sp^3)$ handles in C-C bond-formations. These approaches generally rely on the conversion of traditionally inert chemical functionality into groups susceptible to single-electron activation. Despite this, the α C–C bond activation of ketone derivatives has not yet been fully realized within metallaphotoredox catalysis, but if were so would expand the synthetic chemist´s repertoire for forging $C(sp^3)$ linkages.

In this work we use ketone derived dihydroquinazolinones as radical precursors in metallaphotoredox events, to formally deliver ketone α C−C cleavage driven by the formation of aromaticity via single-electron-oxidation (Fig. 1d)[49–53]. Our strategy allows for abundant ketones to be formally used as cross-coupling synthons with aryl and alkyl bromide electrophiles in the construction of $C(sp^3)$ architectures––currently a need in medicinal chemistry programs[54–57].

## Results

**Optimization of reaction conditions**. We began our investigation by evaluating the reaction of aryl bromide **1** with dihydroquinazolinone **2a** (Table 1), accessed on large scale by the condensation of cyclohexyl methyl ketone with 2-aminobenzamide (2-AB). A combination of Ni(OAc)₂·4H₂O, 4-CzIPN photocatalyst, terpyridine ligand **L4**, Na₂CO₃ and NaBr in NMP under blue light-emitting diodes (LEDs) irradiation at 40 °C provided the best results, affording cross-coupling product **3a** in 93% isolated yield (entry 1). Under the limits of detection, no methyl 4-methylbenzoate arising from $C(sp^3)$–Me cleavage was observed, thus tacitly indicating that C–C cleavage is dictated by the relative stability of the resulting

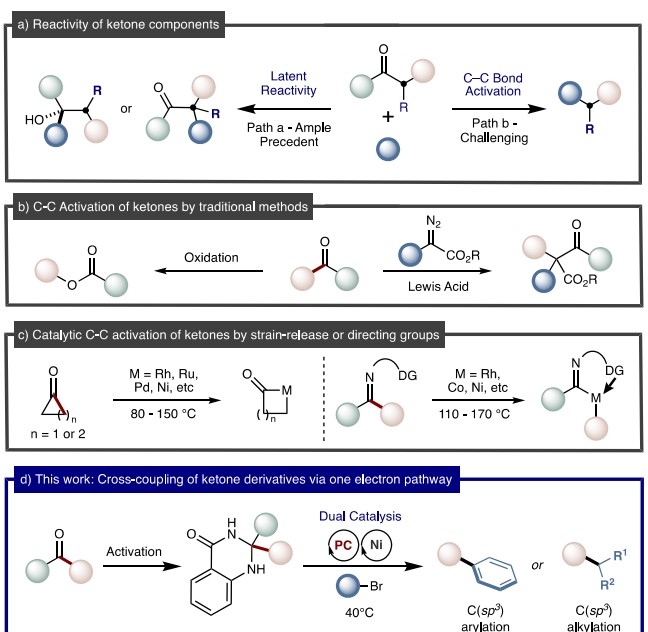

**Fig. 1 C–C bond activation of ketone derivatives. a** Reactivity of ketones. **b** Traditional methods for ketone C–C cleavage. **c** Strategies for catalytic C–C bond activation of ketones. **d** Metallaphotoredox approach for using ketones as cross-coupling synthons.

radical intermediate. As expected, the nature of the ligand played a crucial role. Lower levels of **3a** productivity were attained with 2,2'-bipyridine ligands (entries 2 and 3), while terpyridines other than ligand **L4** were deleterious, highlighting the electronic and steric subtleties of our ligand backbone (entries 4 and 5). Similarly, inferior results were found for nickel pre-catalysts, solvents and bases other than Ni(OAc)₂·4H₂O, NMP and Na₂CO₃ (entries 6–9). Although iridium photocatalysts were competent en route to cross-coupling product **3a** (entry 10), the use of photocatalyst 4-CzIPN constituted a bonus from an accessibility standpoint[58]. As expected, no cross-coupling product (**3a**) was found under the omittance of nickel pre-catalyst, terpyridine ligand **L4**, 4-CzIPN photocatalyst or light (entry 11).

**Substrate scope**. With optimal conditions in hand, we next explored the generality of this $C(sp^3)$ arylation method of ketone derivatives via dihydroquinazolinone activation for C–C bond-cleavage. As shown in Fig. 2 (Left), the $C(sp^3)$ arylation could be accomplished independently on whether dihydroquinazolinones were decorated with primary or secondary alkyl residues. Interestingly, site-selectivity can be easily dictated and modulated by an appropriate selection of the substituents on the dihydroquinazolinone core. Specifically, the coupling of secondary alkyl radicals (**3a–3f**), secondary oxygen-stabilized radicals (**3 g**, **3 h**) or oxygen- or nitrogen-stabilized primary radical congeners (**3i–3l**) could all easily be within reach for Me-substituted analogues. Additionally, it is worth noting that ethyl-substituted dihydroquinazolinones were applicable without deviation in cross-coupling productivity from their methyl-congeners (**3a**, **3l**). The arylation of a primary butyl residue to form **3 m** was found to operate with dihydroquinazolinone cores decorated with methyl groups; note, however, that superior yields were afforded when using aryl-substituted analogues. As such, primary alkyl residues were simply transferred using dihydroquinazolinone cores containing phenyl groups (**3n-r**). It is worth noting this preference for alkyl bond scission over $C(sp^3)$-aromatic cleavage provides an alternative selectivity to transition-metal-catalyzed C–C activations

**Table 1 Optimization of the reaction conditions.**

| Entry | Deviation standard conditions | 3a (%)[a] |
|---|---|---|
| 1 | none | 99 (93)[b] |
| 2 | **L1** instead of **L4** | 36 |
| 3 | **L2** instead of **L4** | <1 |
| 4 | **L3** instead of **L4** | <1 |
| 5 | **L5** instead of **L4** | <1 |
| 6 | Using Ni(COD)₂ | 61 |
| 7 | Using NiCl₂·DME | 92 |
| 8 | using MeCN instead of NMP | 63 |
| 9 | using K₃PO₄ instead of Na₂CO₃ | 81 |
| 10 | using Ir[dF(CF₃)ppy]₂(dtbpy)PF₆ | 69 |
| 11 | no Ni, no **L4**, or in the darkness | <1 |

**1a** (0.10 mmol), **2a** [a](0.12 mmol), Ni(OAc)₂·4H₂O (10 mol%), **L4** (15 mol%), 4-CzIPN (2 mol%), NaBr (1.2 eq.), Na₂CO₃ (1.0 eq.) in NMP (0.10 M) at 40 °C, for 24 h. [b]GC yields using dodecane as standard. [c]Isolated yield.

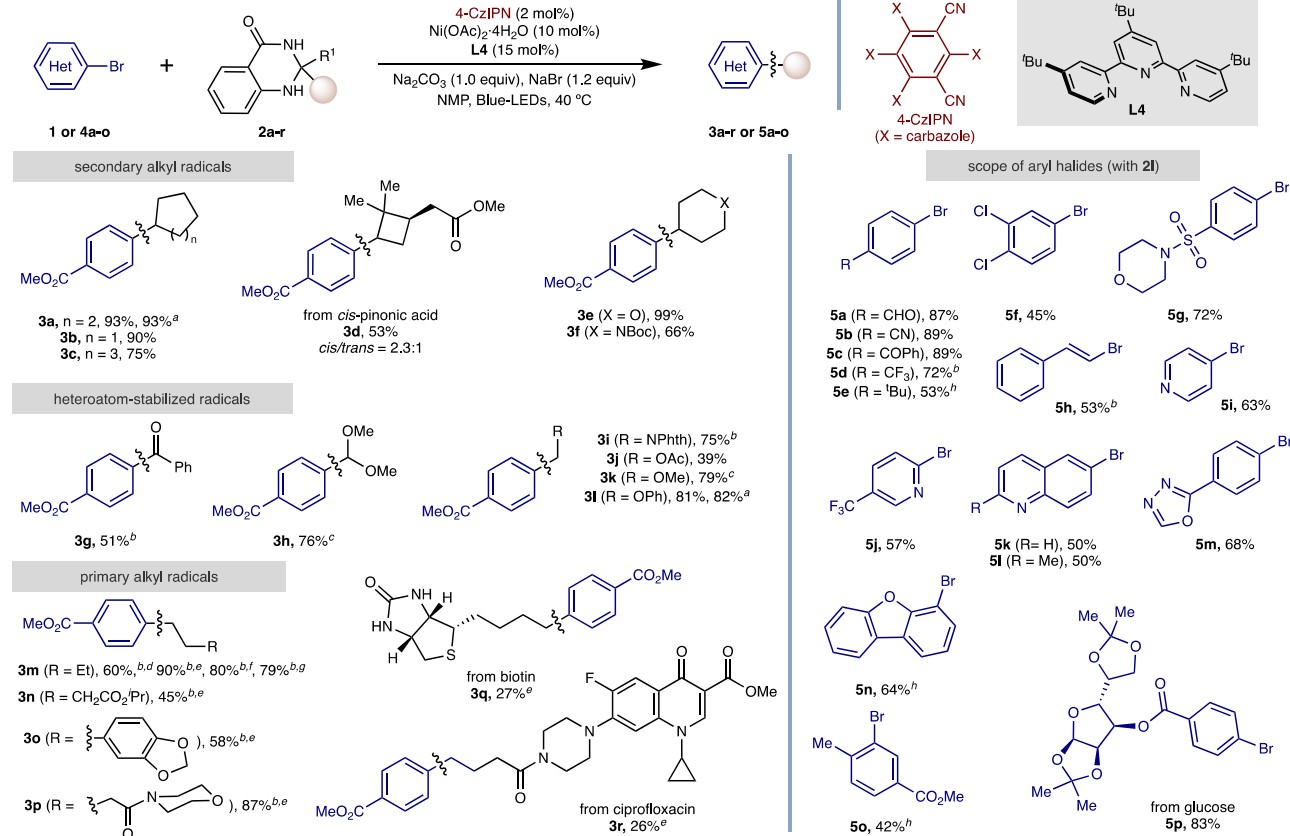

**Fig. 2 Dihydroquinazolinones as *sp³* handles via a C–C cleavage.** As Table 1 (entry 1), using aryl bromide (0.20 mmol). Isolated yields, average of at least two independent runs. Unless stated otherwise, R¹ = Me in the ketone derivative. [a]R¹ = Et in the ketone derivative. [b]Using NiCl₂·DME as Ni source in NMP (0.2 M). [c]Using 5-CzBN (2 mol%) as photocatalyst, LiBr (1.2 eq.) as additive. [d] ¹H NMR yield using CH₂Br₂ as standard. [e]R¹ = Ph in the ketone derivative. [f]R¹ = 4-methoxyphenyl in the ketone derivative. [g]R¹ = benzo[d][1,3]dioxol-5-yl in the ketone derivative. [h]Aryl bromide (0.20 mmol), ketone derivative (0.30 mmol), 4-CzIPN (2 mol%), NiBr₂·diglyme (10 mol%), **L1** (15 mol%), LiHMDS (1.5 eq.), in dioxane (0.1 M).

**Fig. 3 Telescoping the formation of dihydroquinazolinones from ketone congeners.** Using ketone (0.63 mmol) and 2-aminobenzamide (0.6 mmol) to form dihydroquinazolinone, then as Table 1 entry 7 using aryl bromide (0.20 mmol). Yields denote isolated material.

of ketones, which generally give the more stable metal-aryl complex over alkyl species[59–62].

Our dual catalytic platform was found to be widely applicable for an array of aryl bromides regardless of their electronic and steric environment (**5a–5p**) (Fig. 2, Right). As evident from the results compiled in Fig. 2, our method displays an excellent chemoselectivity profile, including accommodation of structures containing aldehyde (**5a**), acetal (**3h**), N-aryl amines (**3r**), thioether (**3q**), amides (**3i**, **3p**, **3q**, **3r**), nitrile (**5b**), ketone (**5c**), sulfonamide (**5g**) and alkyl esters (**3d**, **3j**, **3n**). As shown for cross-coupling products **5h** and **3g**, the reaction could be extended to vinyl bromides or acyl-type radicals with similar ease. Notably, oxygen- and nitrogen-containing heterocycles were compatible in this cross-coupling protocol (**5i–5n**), including pyridine, quinoline, oxadiazole and dibenzofuran scaffolds. For electron-rich or sterically encumbered aryl bromides use of terpyridine ligand **L4** failed to give satisfactory levels of product. For these examples a brief re-optimization found that the use of 2,2'-bipyridine ligand **L1** was effective in exacting the desired transformation (**5e**, **5n**, **5o**). In addition, C($sp^3$) arylation could be affected in the presence of aryl chlorides (**5f**), leaving ample room for further derivatization by other approaches[63,64]. Importantly, our protocol could be employed for accessing biotin (**3q**), ciprofloxacin (**3r**) or glucose (**5p**) containing cross-coupling products, thus showing the prospective potential that this method might have when derivatizing advanced synthetic intermediates. While exploring the substrate scope of this cross-coupling method certain chemical functionalities were found to be incompatible (See Supplementary Section 5.1), including unprotected alcohols, terminal alkynes and dihydroquinazolinones that generate tertiary alkyl radicals. Furthermore, chloride/iodide congeners of **1** and **4h** failed to give satisfactory cross-coupling products whereas the aryl triflate analogue of **1** did give serviceable yields.

In the interest of providing a bonus from an operational standpoint, we wondered whether we could telescope the formation of dihydroquinazolinones from their corresponding ketones. This turned out to be the case with cross-coupling products **3a** and **3l** being obtained in one-pot from ketones **6a** and **6b** in synthetically useful yields via their respective dihydroquinazolinone alkyl radical precursors (**2a**, **2l**) (Fig. 3).

Encouraged by the preceding results, we were interested to see whether our protocol could be extended to the coupling of unactivated alkyl halides. If successful, it would allow dihydroquinazolinones to be utilized as vehicles to forge C($sp^3$)–C($sp^3$) bonds. Exposure of alkyl bromide **7a** to dihydroquinazolinone **2e** under the Ni/**L4** regime used in Fig. 2 failed to provide satisfactory levels of C($sp^3$)–C($sp^3$) bond-formation (**8a**). Gratifyingly, after a brief re-optimization a protocol based on the Ni/**L6** regime turned out to be particularly applicable for the coupling of unactivated alkyl bromides (Fig. 4). As shown, our method was suited not only for the formation of C($sp^3$)–C($sp^3$) linkages arising from the coupling of primary unactivated alkyl halides with secondary alkyl radicals (**8a-k**), but also the coupling of primary alkyl halides with primary alkyl radical species (**8l-8n**). As part of the latter, we used this approach to synthesize the

ethyl-ester of gemfibrozil (**8n**), a medication for dyslipidemia, providing an unconventional disconnection towards this target. In addition, the coupling of secondary alkyl halides with primary alkyl radical intermediates could also be realized, delivering cross-coupling products **8o** and **8p**. Furthermore, alkyl bromides bearing oxygen-, sulfur- and nitrogen-containing heterocycles all successfully participated (**8g-8i**) in the intended C($sp^3$)–C($sp^3$) cross-coupling reaction. More importantly, cross-coupling products **8j**, **8k** and **8p** arising from the conjoining of estrone, cholesterol or oxaprozin containing alkyl bromide derivatives posed no problems, thus holding promise for the application of our protocol when coupling advanced intermediates. Although in comparatively lower yields than those shown in Fig. 2, these results should be benchmarked against the challenge of catalytic C($sp^3$)-C($sp^3$) cross-coupling by offering a complementary technique to existing approaches[65–67].

**Mechanistic studies**. To gain insight into the possible reaction pathway of this cross-coupling process a set of preliminary mechanistic experiments have been carried out (Fig. 5). Firstly, the cross-coupling of aryl bromide **1** with dihydroquinazolinone **2e** was completely inhibited in the presence of a stoichiometric amount of TEMPO radical scavenger, with only the TEMPO-tetrahydropyran adduct (**9**) being observed (Fig. 5a). Subjection of our metallaphotoredox reaction conditions to a cyclopropane containing dihydroquinazolinone (**2t**) yielded only the ring-opened cross-coupling product (**10**) along with quinazolin-4-one by-product (**11**) (Fig. 5b, Top). Furthermore, use of dihydroquinazolinone **2u** gave a mixture of the linear (**12**) and cyclized (**13**) arylation products, which presumably arise from radical 5-exo-trig cyclisation of the intermediary primary hex-1-enyl radical (Fig. 5b, Bottom). The oxidation potential of dihydroquinazolinone **2e** ($E_{1/2}^{ox} = +1.07$ V vs SCE in NMP) was measured using cyclic voltammetry and was shown to be within the oxidizing power of 4-CzIPN ($+1.43$ V vs SCE) (See Supplementary Fig. 21 and 22)[68]. Stern–Volmer fluorescence quenching experiments verified that the excited state of 4-CzIPN was effectively quenched by dihydroquinazolinone **2e** and not by aryl bromide **4d** (See Supplementary Fig. 17). These observations suggest a canonical reductive quenching scenario where single-electron transfer from dihydroquinazolinone to photoexcited 4-CzIPN occurs, initiating formal C–C cleavage en route to alkyl radical driven by the formation of an aromatic by-product.

Terpyridine ligated nickel complex (**Ni-I**) was obtained by exposure of Ni(COD)$_2$/PPh$_3$ to aryl bromide **4d** followed by ligand exchange with terpyridine **L4**[69], with the structure of this complex confirmed by X-ray diffraction. As anticipated, isolated complex **Ni-I** was found to be catalytically competent in the cross-coupling of dihydroquinazolinone **2l** with aryl bromide **4d** (Fig. 5c, Right). Next, we performed the stoichiometric reaction between dihydroquinazolinone **2l** and isolated complex **Ni-I** affording the cross-coupling product **5d** in 25% yield. This suggests that **Ni-I** and similar nickel complexes can capture radicals and undergo reductive elimination under our established conditions (Fig. 5c, left). A positive linear relationship between **Ni-I** catalyst concentration and linear selectivity in the cross-coupling of dihydroquinazolinone **2u** with **4d** was observed (Fig. 6). This is consistent with the formation of C($sp^3$)-centred hex-1-enyl radical from dihydroquinazolinone **2u**, which is captured by **Ni-I**. Higher concentrations of **Ni-I** shortens the lifetime of the alkyl radical in solution resulting in diminished cyclization product **13** formation and greater selectivity for the linear product (**12**).

Based on the aforementioned mechanistic experiments and literature precedent[70], a plausible mechanism was proposed

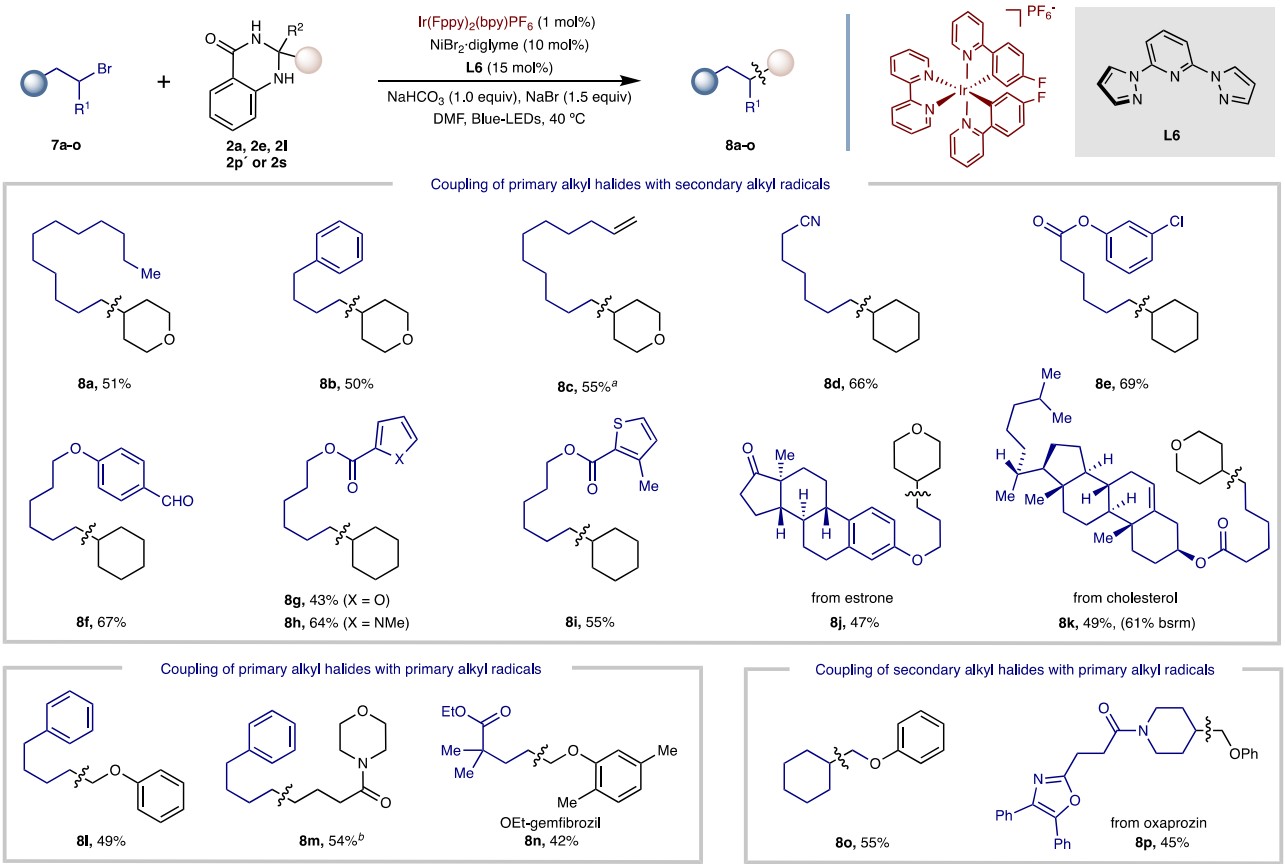

**Fig. 4 Scope of *sp³* alkylation.** Alkyl bromide (0.20 mmol), dihydroquinazolinone (0.30 mmol), NiBr₂·diglyme (10 mol%), **L6** (15 mol%), NaBr (1.5 eq.), NaHCO₃ (1.0 eq.) in DMF (0.1 M) at 40 °C for 24 h. Isolated yields, average of two independent runs. Unless stated otherwise, R² = Me in the ketone derivative. ªDMF (0.2 M). ᵇR² = Ph in the ketone derivative.

(Fig. 7). Oxidative single-electron transfer from dihydroquinazolinone (**I**) to excited photocatalyst triggers a C–C scission driven by the formation of aromatic by-product, forming alkyl radical **II** and reduced photocatalyst. Ni(II) pre-catalyst **III** can be reduced to the Ni(I) form **IV** and then to the catalytically active Ni(0) state **V** by consecutive single electron transfer events with the photocatalyst using dihydroquinazolinone (**I**) as a sacrificial reductant in a catalytic quantity. Oxidative addition of an aryl or alkyl bromide to Ni(0)Lₙ (**V**) generates Ni(II) species (**VI**). Radical recombination of Ni(II) species (**VI**) with alkyl radical **II** generates discrete Ni(III) species **VII**, which upon reductive elimination forms the targeted cross-coupling product (**VIII**) and LₙNi(I)Br **IX**. The two catalytic cycles are then simultaneously closed with a final single-electron transfer between the radical anion of the photocatalyst and LₙNi(I)Br **IX**, recovering both Ni(0)Lₙ **V** and ground-state photoredox catalyst.

## Discussion

In summary, we have developed a catalytic blueprint for forging C(*sp³*)-C(*sp²*) and C(*sp³*)-C(*sp³*) bonds by using ketone derived dihydroquinazolinones as one-electron C(*sp³*) handles via α C–C bond cleavage. This technology offers an unconventional disconnection within the retrosynthetic planning phase of synthesis by enabling C(*sp³*)-arylations and C(*sp³*)-alkylations with an excellent chemoselectivity profile while operating under ambient temperature. In addition, a judicious choice of the starting precursor allows to control the site-selectivity of C–C bond-cleavage. Mechanistic experiments were conducted, all of which are consistent with the operation of a reductive quenching photoredox cycle, beginning with oxidative single-electron transfer of

dihydroquinazolinone radical precursor by excited-state photocatalyst resulting in radical fragmentation driven by formation of an aromatic by-product. Further extensions to other related processes are underway in our laboratories.

## Methods

**General procedure for nickel-catalyzed coupling with aryl bromides**. An oven-dried 8 mL screw-cap test tube containing a stirring bar was charged with 4-CzIPN (3.2 mg, 2 mol%), Ni(OAc)₂·4H₂O (5.0 mg, 10 mol%), 4,4',4"-tri-*tert*-butyl-2,2':6'2"- terpyridine (12.1 mg, 15 mol%), NaBr (24.7 mg, 1.2 eq.), aryl bromide **1** (if solid, 1.0 eq., 0.2 mmol) and ketone derivative **2** (1.2 eq.). The test tube was introduced in a nitrogen-filled glovebox where Na₂CO₃ (21.2 mg, 1.0 eq.) was added. The reaction vessel was sealed with a screw cap and removed from the glovebox. Afterwards, aryl bromide **1** (if liquid) and NMP (2 mL, 0.1 M) were added by syringe. Parafilm was used to reseal the pierced cap. The reaction mixture was stirred at rt for 1 min, then exposed to blue LED irradiation at 40 °C for 24 hours. The reaction mixture was quenched with water/brine (10 mL) and extracted with ethyl acetate (3 × 10 mL). The combined organic extracts were dried (Na₂SO₄), concentrated under reduced pressure and purified by silica gel chromatography to afford the desired product **3** or **5**.

**General procedure for nickel-catalyzed coupling with alkyl bromides**. An oven-dried 8 mL screw-cap test tube containing a stirring bar was charged with Ir(Fppy)₂(bpy)PF₆ (1.6 mg, 1 mol%), 2,6-di(1-pyrazolyl)pyridine (12.1 mg, 15 mol%), NaBr (30.9 mg, 1.5 eq.), alkyl bromide **6** (if solid, 1.0 eq., 0.2 mmol) and ketone derivative **2** (1.5 eq.). The test tube was introduced in a nitrogen-filled glovebox where NiBr₂·diglyme (7.1 mg, 10 mol%) and NaHCO₃ (16.8 mg, 1.0 eq.) were added to the reaction vessel. The reaction tube was sealed with a screw cap and removed from the glovebox. Afterwards, alkyl bromide **6** (if liquid) and DMF (2 mL, 0.1 M) were added by syringe. Parafilm was used to reseal the pierced cap. The reaction mixture was stirred at rt for 1 min, then exposed to blue LED irradiation at 40 °C for 24 hours. The reaction mixture was quenched with water/brine (10 mL) and extracted with ethyl acetate (3 × 10 mL). The combined organic extracts were dried (Na₂SO₄), concentrated under reduced pressure and purified by silica gel chromatography to afford the desired product **8**.

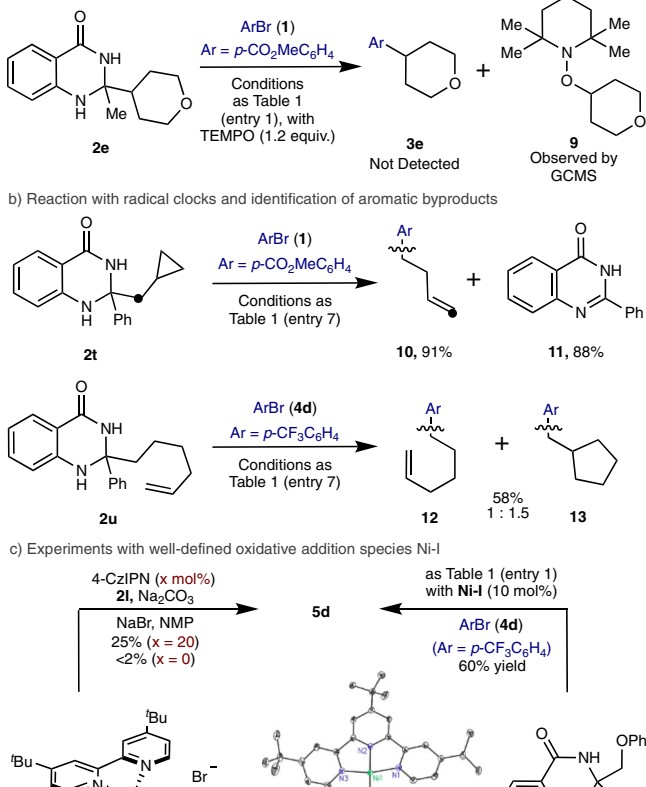

**Fig. 5 Preliminary mechanistic experiments. a** TEMPO radical trapping studies. **b** Radical clock experiments. **c** Mechanistic experiments with well-defined nickel species.

**Fig. 7 Proposed mechanism.** Proposed reaction pathway involves a reductive quenching photoredox cycle for the generation of alkyl radical from dihydroquinazolinone, which is captured by Ni(II) oxidative addition complexes to form Ni(III) species for subsequent cross-coupling by reductive elimination. The photoredox and nickel catalytic cycles are simultaneously closed by electron transfer from reduced photocatalyst to Ni(I) species generated post-reductive elimination.

Cambridge Crystallographic Data Centre. Any further relevant data are available from the authors on request.

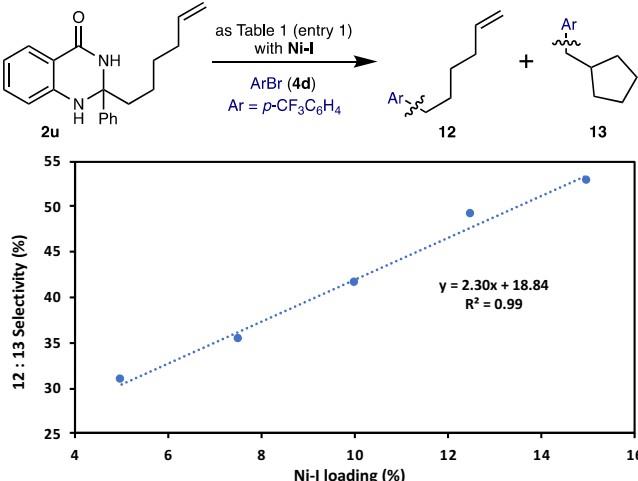

**Fig. 6 Radical cyclization as a function of catalyst loading.**
**2 u** (0.12 mmol), **4d** (0.10 mmol), Ni(OAc)$_2$·4H$_2$O (10 mol%), **L4** (15 mol %), 4-CzIPN (2 mol%), NaBr (1.2 eq.), Na$_2$CO$_3$ (1.0 eq.) in NMP (0.10 M) at 40 °C, for 24 h.

## Data availability
The data supporting the findings of this study are available within the article and its Supplementary Information file. CCDC 2102869 (**Ni-1**) contain the supplementary crystallographic data for this paper. These data can be obtained free of charge from The

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

## Acknowledgements

We thank ICIQ, FEDER/MCI−AEI/PGC2018-096839-B-I00, MCIN/AEI/10.13039/501100011033 (CEX2019-000925-S) and European Research Council (ERC) under

European Union's Horizon 2020 research and innovation program (grant agreement No 883756) for financial support. X. L. thanks the China Scholarship Council (CSC) for a predoctoral fellowship.

## Author contributions

R.M. and X.-Y.L. conceived and designed the study. X.-Y.L. and R.A. performed the experiments and collected the data. R.M. and R.A. wrote the manuscript and all authors commented on the paper.

## Competing interests

The authors declare no competing interests.
