## [Peer Review File · Nature Communications]

REVIEWER COMMENTS

Reviewer #1 (Remarks to the Author):

Martin and co-workers reported a method to utilize the dihydroquinazolinones derives of ketones to forge C-C bond via dual catalysis system. The C(sp²)-C(sp³) and C(sp³)-C(sp³) bonds can be efficiently constructed with this method. This work provides a new way to cleavage the α C-C bond of ketones. Although the authors always emphasize the transformation of ketones, they didn't give clear feasibility on the ketone types. And the work needs further improvement not only on the substrate scopes but also the application of this method. While this paper appears to have been convincingly carried out and with good novelty, unfortunately, the reviewer thinks the current form lacks systematic studies at this moment and dose not reach the high standards to warrant publication in Nature Communication. This manuscript could be considered if the authors address these issues.

Page3 Notably, the presence of oxygen- and nitrogen-.....(5i-5l). The latter isto the nickel center. So 5l? or maybe means 5j? How about the pyridine or quinoline substrates without nitrogen α substituents?

How about aryl iodides or aryl pseudo-halides? Only aryl bromides were tested under the conditions.

The authors declare that they provide a method to unravel the ketone derivatives as traceless synthetic handles, however, it seems that only methyl ketones and phenyl ketones can be utilized. How about other type of the ketones? If the both sides are alkyl groups (nonmethyl group), how is the selectivity? The author should make it clear in the manuscript.

Identically, for the tert-alkyl methyl (or alkyl) ketones or tert-alkyl phenyl ketones, what are the results? They can not be employed or not. The authors should give more clear information for the adaptation and limitation of this method.

An intramolecular cyclization reaction with a terminal alkene on the mechanism investigation should be considered. The linear relationship between the ratio of normal coupling product to the cyclization product and the nickel concentration could give more insight to the "open-shell species" as the author mentioned (see: JACS 2019, 141, 6726).

The deeper mechanistic studies (experiments and DFT calculations) need be conducted to understand the mechanism and its limitation.

Application of this method in synthesis to show its advantage should be considered by the authors.

In addition, the language needs to be thoroughly checked. Some sentences are super-long (eg. abstract)

Reviewer #2 (Remarks to the Author):

The authors report an efficient metallaphotoredox catalyzed cross coupling of dihydroquinazolinones with aryl and alkyl bromides via C-C bond cleavage. The use of alcohols, primary amines, and aldehydes as alkylation reagent have been studied extensively in C-C bond formation, but ketone derivatives have not yet been realized in this transformation. Furthermore, the authors further demonstrated the coupling of aryl and alkyl bromides with diverse ketone

derivatives through merging with nickel catalysis, which greatly expanded the scope of alkylating partners. The reaction scope is broad (47 examples) and a wide variety of functional groups are tolerated. I highly recommend its publication after addressing some minor comments listed below:

1. In figure 3 and Figure 4, the footnote with scheme should be located in one page.
2. How about vinyl halides in this system?

Reviewer #3 (Remarks to the Author):

The manuscript by Martin and co workers describes a method for the coupling of alkyl radicals (generated formally from ketones) from N,N-linked dihydroquinazolines. The key to the success is the merger of photocatalysis and Ni-catalysis. The photocatalytic cycle generates the alkyl radicals via scission of the DHQs and the Ni-cycle couples the aryl group via pathways first introduced by MacMillan and Molander.

The reaction shows good scope in the alkyl fragment and aryl fragment and importantly, can be extended to coupling with alkyl bromides (leading to sp³-sp³ coupling). A selection of mechanistic experiments enables the proposition of a plausible catalytic cycle.

There isn't too much to add here. It's a very nice paper highlighting an interesting solution to a persistent problem, namely sp²-sp³ and sp³-sp³ cross coupling. The yields are good, scope is broad and the reaction seems operationally straightforward. Therefore, I think it would make an excellent addition to Nat. Comm.

There are two aspects I would suggest improvement on

1. The schematical representation is really quite hard to follow. I found the bewildering number of coloured spheres, black dots and R groups almost incomprehensible. I would recommend the authors rethink how they display their results to maximise clarity

2. It would be instructive to show in the catalytic cycle the entry step of the Ni(II) precatalyst. There doesn't need to be anything fancy or complicated here. Just the generation of Ni(0) from the Ni(II) catalyst. Ni(0) is clearly the active species, but Ni(II) is the added salt. While I accept this is well documented, I think the reader would benefit from this clarity

otherwise

accept

→ Reviewer 1

Although this reviewer has recognized the significance and novelty of our work by providing a new way to cleave the α C-C bond of ketones, he/she argues that the work needs further improvement. We truly appreciate the constructive comments and suggestions received by this reviewer, as it allowed us to carry out some additional experiments that, in our opinion, make our manuscript more synthetically appealing for a broader audience.

All the reviewer comments have been addressed:

- **Point 1:** Page 3. Notably, the presence of oxygen- and nitrogen-.... (5i-5l). The latter is.... To the nickel center. So 5l ? or maybe means 5j ?. How about the pyridine or quinoline substrates without nitrogen α -substituents ?

Response: This was a very insightful observation from the referee that we initially failed to appreciate. Specifically, our examples of using pyridyl and quinolinyl bromides (**5j** and **5l**) both contained substituents *ortho* to the nitrogen of these heterocycles. This may in turn attenuate competitive binding of these substrates to nickel centers over the ligand, and so avoid catalyst deactivation. Thus, these substrates may be seen as privileged and not generally representative of pyridyl and quinolinyl bromides. To answer this query, we tested pyridyl and quinolinyl bromides without a substituent *ortho* to the nitrogen, and found that both performed well under our optimized reaction conditions. These results have been incorporated into the manuscript (See Fig. 2, **5i** and **5k**).

- **Point 2:** : How about aryl iodides or aryl pseudo-halides? Only aryl bromides were tested under the conditions ?

Response: We totally agree with the reviewer's curiosity in whether aryl halides or pseudo-halides outside of aryl bromides are amenable in this method. If possible, it would greatly expand the potential substrates that can partake in this methodology. Unfortunately, methyl 4-iodobenzoate failed to give cross-coupling product, while methyl 4-chlorobenzoate gave trace product formation. In contrast, we gratifyingly found the triflate analogue did give modest levels of productivity without deviation from our standard reaction conditions. These results have been incorporated into the supporting information (Section 5.1), along with a sentence in the text of the paper briefly summarising these results.

- Point 3:** The authors declare that they provide a method to unravel the ketone derivatives as traceless synthetic handles, however, it seems that only methyl ketones and phenyl ketones can be utilized. How about other type of the ketones? If the both sides are alkyl groups (nonmethyl group), how is the selectivity? The author should make it clear in the manuscript

Response: The reviewer makes a great point and we do concede that we had only attempted to use dihydroquinazolinone cores containing methyl and phenyl groups, mainly because of the availability of the ketone precursors. However, we do agree it would be very informative to the reader to know how other ketone cross-coupling synthons outside of acetyl and benzoyl groups would perform. Towards this end we prepared dihydroquinazolinone radical precursors that contain ethyl dummy groups (i.e. non-methyl). For dihydroquinazolinones bearing an ethyl and a cyclohexyl group only cross-coupling of the cyclohexyl was observed, demonstrating selective C-C scission of secondary alkyl groups over primary alkyl groups.

For dihydroquinazolinones bearing an ethyl and a $-\text{CH}_2\text{OPh}$ unit only cross-coupling of the $-\text{CH}_2\text{OPh}$ group was observed, showing selective C-C scission of hetero-atom stabilized primary alkyl groups versus primary alkyl groups.

For dihydroquinazolinones bearing a methyl group and a *n*-butyl group only cross-coupling of the *n*-butyl group was observed, demonstrating selective C-C scission of primary alkyl groups over methyl. However, for the cross-coupling of primary alkyl fragments superior yields were obtained with dihydroquinazolinone radical precursors containing a phenyl dummy group. With the preceding in mind we briefly explored how the phenyl dummy group of a dihydroquinazolinone radical precursor could be functionalized and found that 4-methoxyphenyl and benzo[*d*][1,3]dioxol-5-yl groups performed well. These results have been added to the manuscript (See Fig. 2) and the supporting information.

- Point 4:** Identically, for the tert-alkyl methyl (or alkyl) ketones or tert-alkyl phenyl ketones, what are the results? They cannot be employed or not.

Response: The reviewer makes a good point that we demonstrate primary and secondary alkyl cross-coupling partners to be competent, so it would naturally beg the question of would the same be true for tertiary alkyl partners. We prepared the necessary radical precursor, but when used under our conditions with aryl bromide **1** no cross-coupling product was observed. This may be unsurprising as the use of tertiary radicals is well understood to be problematic in metallaphotoredox settings with nickel, as radical-metal recombination is slow under inner-sphere mechanisms.

While considering the previous experiment, we did attempt to adapt conditions from Molander and Primer (*J. Am. Chem. Soc.* **2017**, *139*, 9847–9850) who reported a method of incorporating tertiary alkyl radicals into metallaphotoredox manifolds. This approach did yield some cross-coupling productivity, but in low amounts. These results have been added to the supporting information (See Section 5.1).

- Point 5:** The authors should give more clear information for the adaptation and limitation of this method.

Response: We whole heartedly agree with the reviewer that dissemination of negative outcomes can be just as informative as positive results for any potential users of this method. To create a more informative piece of work we have created a list of unsuccessful dihydroquinazolinone radical precursors, aryl halides and alkyl bromides used (See Supporting Information, Section 5.1). We then briefly summarize these limitations in the text of the paper.

- Point 6:** An intramolecular cyclization reaction with a terminal alkene on the mechanism investigation should be considered. The linear relationship between the ratio of normal

coupling product to the cyclization product and the nickel concentration could give more insight to the “open-shell species” as the author mentioned (see: JACS 2019, 141, 6726).

Response: We completely concur with the reviewer that the ratio of cyclized to linear cross-coupling products for a hex-1-enyl radical would be informative of the rate at which the alkyl radical combines with the nickel catalyst. As such, we prepared the appropriate radical precursor (**2u**) to generate a hex-1-enyl radical, which when subjected to our standard cross-coupling conditions with aryl bromide **4d** gave approximately a 0.4:0.6 ratio of linear to cyclized product. Implying the rate of union of the hex-1-enyl radical with nickel is somewhat similar to the rate of cyclisation.

We also performed the recommended series of experiments to assess how the linear/cyclized selectivity changes with catalyst concentration. Here we used our standard reaction conditions while using the oxidative addition complex **Ni-I** to ensure consistency of the concentration of active nickel species. We observed a positive linear relationship between **Ni-I** catalyst concentration and linear selectivity in the cross-coupling of dihydroquinazolinone **2u** with aryl bromide **4d**. This is consistent with the formation of C(sp³)-centered hex-1-enyl radical from dihydroquinazolinone **2u**, which is captured by **Ni-I**. Higher concentrations of **Ni-I** shortens the lifetime of the alkyl radical in solution resulting in diminished cyclized product **13** formation and greater selectivity for the linear product (**12**). These results have been added to the manuscript and supporting information (See Fig 5. And Fig 6.).

- **Point 7:** The deeper mechanistic studies (experiments and DFT calculations) need be conducted to understand the mechanism and its limitation.

Response: We totally agree with the reviewer that DFT calculations might aid understanding this process a bit better to aid future developments of this method. However, we do feel the suite of preliminary mechanistic experiments we have conducted (including TEMPO radical probes, radical clocks, fluorescence quenching, cyclic voltammetry, nickel-complex studies) make for a good initial foray towards mechanistic understanding of this process to accompany the report of its synthetic utility.

- **Point 8:** Application of this method in synthesis to show its advantage should be considered by the authors.

Response: We totally agree with the reviewer that any new synthetic methodology should attempt to be used in the preparation of known targets as a benchmark to compare with other synthetic approaches. We used our C(sp³)-C(sp³) method to synthesize the ethyl-ester of gemfibrozil (**8n**), a medication for dyslipidemia, providing a new and unconventional disconnection towards this target. This result has been added to the manuscript and supporting information (See Fig. 4).

- **Point 9:** In addition, the language needs to be thoroughly checked. Some sentences are super-long (eg. abstract)

Response: After reviewing the language of our manuscript we concede that some sentences were quite long. We have made a particular effort to shorten long sentences and enhance the clarity of our writing.

→ Reviewer 2

This reviewer has clearly recognized the value of our work by highlighting the following:

“The use of alcohols, primary amines and aldehydes as alkylation reagents have been extensively studied in C-C bond-formation, but ketone derivatives have not yet realized in this transformation. [...] I highly recommend its publication after addressing some minor comments”

All the reviewer comments have been addressed:

- **Point 1:** In figure 3 and Figure 4, the footnote with scheme should be located in one page.

Response: We thank the reviewer for pointing out this schematic error. We have redoubled our efforts to ensure all figures occupy the same page as their footnotes

- **Point 2:** How about vinyl halides in this system?

Response: We shared the reviewer’s curiosity in how different vinyl halides might affect the outcome of our cross-coupling method. We found that while a vinyl bromide gave good yields, the analogous iodide failed to give cross-coupling product and chloride gave a diminished yield. These results have been added to the supporting information (See Section 5.1).

→ Reviewer 3

This reviewer has clearly recognized the value of our work by highlighting the following:

“There isnt too much to add here. Its a very nice paper highlighting an interesting solution to a persistant problem, namely sp²-sp³ and sp³-sp³ cross coupling. The yields are good, scope is broad and the reaction seems operationally straightforward. Therefore, i think it would make an excellent addition to Nat. Comm.”

All the reviewer comments have been addressed:

- **Point 1:** The schematical representation is really quite hard to follow. I found the bewildering number of coloured spehere, black dots and R groups almost incomprehensible. I would recommend the authors rethink how they display their results to maximise clarity.

Response: We thank the reviewer for bringing to our attention that the graphical representation of this work was hard to follow. We have taken this comment seriously and edited all figures to reduce the number of design elements and have streamlined our footnotes. We hope this will give greater clarity to the results of our work.

- **Point 2:** it would be instructive to show in the catalytic cycle the entry step of the Ni(II) precatalyst. There doesn't need to be anything fancy or complicated here. Just the generation of Ni(0) from the Ni(II) catalyst. Ni(0) is clearly the active species, but Ni(II) is the added salt. While I accept this is well documented, I think the reader would benefit from this clarity?

Response: We whole heartedly agree with the reviewer that this subtle mechanistic assumption may cause confusion to the broader readership of Nature Communications. We have added to our proposed mechanism (Fig. 7) a route in where Ni(II) pre-catalyst can be converted into the catalytically active Ni(0) form by sequential single electron reductions from the photocatalyst, by sacrificing a catalytic quantity of dihydroquinazolinone radical precursor as reductant

REVIEWERS' COMMENTS

Reviewer #1 (Remarks to the Author):

The authors have addressed all the questions raised by the referees. Thus, the paper could be published as it is , no further changes are required.